# CODE REPRESENTATION LEARNING AT SCALE

**Dejiao Zhang**[*] **& Wasi Uddin Ahmad**[*]
{dejiaoz,wuahmad}@amazon.com

**Ming Tan & Hantian Ding**
{mingtan,dhantian}@amazon.com

**Ramesh Nallapati & Dan Roth & Xiaofei Ma & Bing Xiang**
{rnallapa,drot,xiaofeim,bxiang}@amazon.com

**AWS AI Labs**

ABSTRACT

Recent studies have shown that code language models at scale demonstrate significant performance gains on downstream tasks, *i.e.,* code generation. However, most of the existing works on code representation learning train models at a hundred million parameter scale using very limited pretraining corpora. In this work, we fuel code representation learning with a vast amount of code data via a two-stage pretraining scheme. We first train the encoders via a mix that leverages both randomness in masking language modeling and implicit structure and semantic aspects of programming language. We then enhance the representations via contrastive learning with hard negative and hard positive constructed in an unsupervised manner. We establish an off-the-shelf encoder model that persistently outperforms the existing models on a wide variety of downstream tasks. To comprehend the factors contributing to successful code representation learning, we conduct detailed ablations and share our findings on *(i)* a customized and effective token-level denoising scheme for source code; *(ii)* the importance of hard negatives and hard positives; *(iii)* how the proposed bimodal contrastive learning boost the cross-lingual semantic search performance; and *(iv)* how the pretraining schemes decide the downstream task performance scales with the model size. [1]

## 1 INTRODUCTION

Large language models (LLMs) pretrained on a massive amount of source code have reshaped the landscape of code generation (Chen et al., 2021; Chowdhery et al., 2022; Li et al., 2023, *inter alia*). As an example, the recent release of a 6TB dataset (Kocetkov et al., 2022) comprising source code under permissive licenses play pivotal roles in promoting the advancement of code language models in present times. Nonetheless, these large corpora are not fully utilized to develop general-purpose Programming Language (PL) embedding models. To date, most PL embedding models (Feng et al., 2020a; Guo et al., 2021; 2022, *inter alia*) have no more than 125M parameters and are primarily trained on a few millions of training examples, *e.g.,* CodeSearchNet (Husain et al., 2019).

Despite the undeniable significance of large-scale data, it's imperative to acknowledge the vital role of pretraining objectives. The prevailing approach for pretraining a bidirectional Transformer encoder to learn representations is through the optimization of a masked language modeling (MLM) objective, as proposed by Devlin et al. (2019b). The masking scheme in the standard MLM objective follows an 80-10-10 practice.[2] However, we have noticed that such a masking scheme leads to the development of suboptimal code embedding models. Since code snippets contain both natural language (NL) statements (*i.e.,* docstrings, comments) and pure code, hence replacing masked tokens with a random token following the 80-10-10 convention could result in replacing an NL token with a PL token, and vice versa (see statistics in Appendix A.3). We speculate such co-occurrence of PL and NL together with the syntax nature of source code make it easier to disrupt both the semantics and structure of the masked code, resulting in sub-optimal learning of the language model.

---

[*]Corresponding authors with equal Contribution.

[1]Code and models can be found at https://code-representation-learning.github.io/.

[2]Under this scheme, 80% of the randomly selected tokens for prediction are replaced with the [MASK] token, 10% are substituted with random tokens, and the remaining tokens remain unchanged.

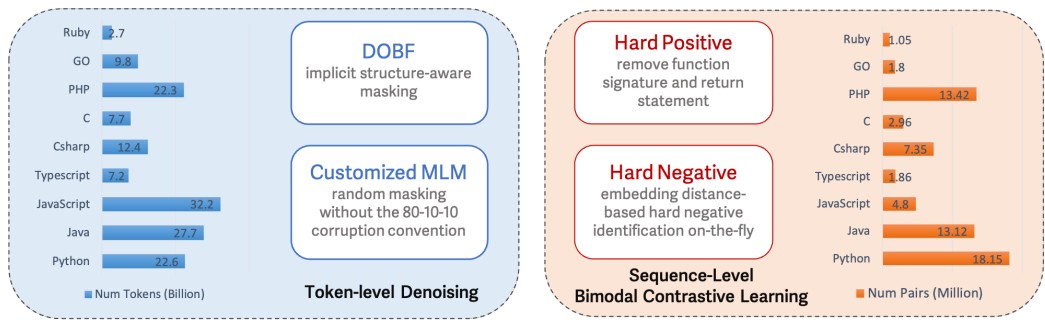

Figure 1: An overview of the key ingredients of CODESAGE for code representation learning.

While MLM pretraining yields contextual token representations, most downstream discriminative tasks primarily function at the sequence level. When the objective is to enhance the representation discrimination power for immediate application in sequence-level tasks, contrastive learning (CL) emerges as the go-to approach. Existing works have employed unimodal CL (using Code-Code pairs) (Guo et al., 2022; Jain et al., 2021) or bimodal CL (using Text-Code pairs) (Li et al., 2022) for representation learning. In unimodal CL, a popular choice is to utilize *dropout* augmentation Gao et al. (2021) to construct positive code pairs. However, we found that *dropout* augmentation suffers from supporting long training process, also reported by Zhou et al. (2022). In contrast, bimodal CL becomes an appealing choice, primarily because of the availability of naturally occurring pairs. Prior studies utilize functions and their corresponding docstrings to establish the bimodal training pairs. Nonetheless, our preliminary experiments indicate that substantial overlap between docstrings and function signatures simplifies the contrastive learning process (see statistics in Appendix A.6).

To this end, we present CODESAGE, a bidirectional encoder representation model for source code. We pretrain CODESAGE using a two-stage training scheme with a large amount of customized pre-training data (Kocetkov et al., 2022). We depict the key ingredients of CODESAGE in Figure 1. We first train the bidirectional encoders via a mix of two objectives complementing each other: identifier deobfuscation (DOBF) and MLM without the 80-10-10 practice. Similar to a human programmer, finding meaningful names for obfuscated identifiers necessitates the model to acquire a profound comprehension of code semantics and structure. Meanwhile, as a more general objective, MLM covers other facets beyond identifiers of code – this is important for enriching the training signals, especially for data examples with non-informative identifier names. In the second stage, we leverage the *(text, code)* pairs for bimodal contrastive learning (CL). In contrast to existing approaches that primarily rely on naturally occurring text and code pairs, we propose a strategy to reduce the likelihood of the model learning shortcuts. Our approach involves exclusively utilizing the function body while disregarding the signature and return statements. We additionally harness CL based on hard negatives identified within the embedding space. We show that such a hard positive and negative construction strategy is simple, yet essential for effective bimodal contrastive learning.

We train three bidirectional encoder representation models, namely, CODESAGE-SMALL (130M), CODESAGE-BASE (356M), and CODESAGE-LARGE (1.3B). We assess the effectiveness of our approach over a wide variety of discriminative tasks, where CODESAGE substantially outperforms the previous state-of-the-art models with similar model sizes on most tasks. To comprehend the factors contributing to successful code representation learning, we meticulously analyze the key components of our framework and present our findings for future research endeavors.

## 2 RELATED WORKS

**Embedding for Programming Languages**   Recently, there has been a surge of interest in learning general-purpose representations to support a wide variety of downstream tasks in programming languages. Feng et al. (2020a); Kanade et al. (2020); Li et al. (2023) take the inspiration of the success in text and optimize the Masking Language Modeling (MLM) objective on the linearized code data. Similar to text, they additionally optimize with replaced token detection objective (Clark et al., 2020) or the next sentence prediction objective (Devlin et al., 2019b) for source code. Another line

of work leverages the structure aspect of code to provide additional training signals. Among them, Guo et al. (2021) leverages the data flow to encode the relation of "where-the-value-comes-from" between variables. Wang et al. (2021a); Jiang et al. (2021) inject syntactical structure from the abstract syntax tree (AST) through variant auxiliary objectives. A more recent work (Guo et al., 2022) flattens the AST structure into a sequence directly and encodes the syntax information via language modeling objectives. Wang et al. (2021b); anne Lachaux et al. (2021) train a sequence-to-sequence language model to reconstruct the original code from an identifier-obfuscated code where class, function, and variable names are replaced with special tokens. Deobfuscation implicitly encodes data flow and AST without involving auxiliary objectives or complex input with deep hierarchy, since the model needs to understand the dependency between variables as well as code structure so as to correctly predict the names for identifiers.

**Contrastive Learning** Ever since the early success attained by the Siamese (Hadsell et al., 2006) network, contrastive learning has been widely adopted in representation learning using deep neural networks. Song et al. (2016) extends the vanilla triplet loss by contrasting each positive example against all in-batch negatives, which has greatly improved the learning efficiency and is further popularized by SimCLR (Chen et al., 2020). However, different from the compute version domain where effective positives can be obtained by stochastic transformations of images in the input space, effective data augmentation has long been a challenge in NLP due to the discrete nature of the input. Such challenge is further validated in Gao et al. (2021) which shows that *dropout* (Srivastava et al., 2014) as the minimum data augmentation is often more effective than those obtained by operating in the discrete input space, *e.g.,* word deletion and replacement.

Alternatively, various methods have been proposed to leverage naturally occurring pairs as positives. Zhou et al. (2022) treat the consecutive utterances from dialogue data as positives, while Neelakantan et al. (2022) consider the neighboring texts mined from the internet. A very recent work (Wang et al., 2022) leverages the question and answer or comment pairs from StackExchange and Reddit. In a similar vein for programming language, Guo et al. (2022); Wang et al. (2021a); Neelakantan et al. (2022) leverage (text, code) pairs with text mined from the docstrings. We take a step further by focusing on hard positive and hard negative construction, which is a key ingredient for representation learning and allows us to attain off-the-shelf embedding models.

## 3 METHOD

### 3.1 MASK LANGUAGE MODELING AND DEOBFUSCATION PRE-TRAINING

Given an input sequence with $N$ tokens, *i.e.,* $\mathbf{x} = [\mathbf{x}_1, \mathbf{x}_2, \ldots, \mathbf{x}_N,]$, the mask language modeling objective (Devlin et al., 2019b) is formed as follows

$$\mathcal{L}_{\text{MLM}}(\mathbf{x}) = - \sum_{i \in \mathcal{M}} \log \mathbb{P}\left(\mathbf{x}_i | \mathbf{x}^{\mathcal{M}}\right) \tag{1}$$

Here $\mathcal{M}$ denotes the mask applied on the given input $\mathbf{x}$. Equation (1) is essentially a denoising objective with the task to predict the original tokens given the masked sequence $\mathbf{x}^{\mathcal{M}}$.

**Deobfuscation** We first consider identifier deobfuscation (DOBF) which pretrains the model to predict the masked-out names of the identifiers. Similar to human programmers, in order to deobfuscate the code, the model needs to understand both the semantics and structure of the code. Also the NL tokens, *i.e.,* docstring and comment, are excluded from code obfuscation. When the model is trained to predict the identifier names, it can benefit from looking at and correlating with the NL tokens in comments or docstrings as those often carry rich semantics of code. Consequently, the model is encouraged to learn improved shared representations between PL and NL, as indicated by the better NL2Code search performance attained by DOBF than random masking in Table 3.

DOBF is initially proposed for Seq2Seq models (anne Lachaux et al., 2021; Wang et al., 2021b). To the best of our knowledge, we are the first to apply it to the encoder-only models. The main challenge to adopting DOBF for encoder-only models is to construct the one-on-one mapping between mask tokens (inputs to the LM) and identifier tokens (output labels) due to the differences in code tokenization (*i.e.,* using *tree-sitter*) and model-specific tokenization (*i.e.,* using a *sentencepiece* tokenizer). We briefly discuss the challenge in Appendix A.5.

**Random Masking**    Additionally, we also involve the random token masking strategy in BERT Devlin et al. (2019b) for two main reasons. First, to promote better representations by promoting the model to learn beyond identifiers. Taking Python as an example, there are approximately 30% of the code tokens associated with identifiers, hence better representations can be attained by encoding the information carried by the remaining 70% of tokens. Second, not every programmer follows the naming conventions, *e.g.,* meaningless variable names like $v1, v2, v3$ can be used. Predicting such tokens is unnecessarily hard and provides a very limited training signal.

We do not follow the 80-10-10 masking convention proposed in the standard MLM for text (Devlin et al., 2019b). Since source codes are composed of NL and PL tokens (*i.e.,* identifiers, keywords, operators), random replacement of tokens could hurt both the structure and meaning of code and leads to deterioration in representation learning.[3] We show in Section 4.2.1 that the 80-10-10 convention consistently results in worse performance on downstream tasks. In this paper, we also set the random masking rate to 15% which we find is optimal through our ablation study in Appendix A.4. For each training example, we randomly pick DOBF or random masking with equal probability.

## 3.2    BIMODAL CONTRASTIVE LEARNING WITH HARD NEGATIVE AND HARD POSITIVE

Let $\mathbf{x}_i, \mathbf{x}_{i+}$ denote a positive input pair and $\mathbf{h}_i, \mathbf{h}_{i+}$ be the associated representations output by the last hidden layer of the encoder. Let $\mathcal{B} = \{\mathbf{h}_1, \mathbf{h}_{1+}, \mathbf{h}_2, \mathbf{h}_{2+}, \dots, \mathbf{h}_N, \mathbf{h}_{N+}\}$ denote the representations of a randomly sampled batch with $N$ pairs, we then minimize the following symmetric loss,

$$
\mathcal{L}_{\mathrm{CL}}\left(\mathbf{h}_i, \mathbf{h}_{i+}\right) = - \left( \log \frac{\exp(\mathbf{h}_i \diamond \mathbf{h}_{i+}/\tau)}{\exp(\mathbf{h}_i \diamond \mathbf{h}_{i+}/\tau) + \sum_{k \in \mathcal{B} \backslash (i, i+)} \gamma_i^k \cdot \exp(\mathbf{h}_i \diamond \mathbf{h}_k/\tau)} \right.
$$
$$
\left. + \log \frac{\exp(\mathbf{h}_{i+} \diamond \mathbf{h}_i/\tau)}{\exp(\mathbf{h}_{i+} \diamond \mathbf{h}_i/\tau) + \sum_{k \in \mathcal{B} \backslash (i, i+)} \gamma_{i+}^k \cdot \exp(\mathbf{h}_{i+} \diamond \mathbf{h}_k/\tau)} \right) . \tag{2}
$$

Here, $\tau$ is the temperature hyper-parameter which we set as 0.05 in this work. $\diamond$ denotes cosine similarity between two representation vectors. $\gamma_i^k$ is the weight parameter which we will detail next.

**Hard Negative**    Without supervision, it is tricky to identify hard negatives. We resort to a distance-based unsupervised approximation of hard negatives proposed in Zhang et al. (2021). For a given anchor $\mathbf{h}_i$, hard negatives refer to those semantically different examples but are mapped close to $\mathbf{h}_i$ in the representation space. Thereby, the closer a negative is to the anchor $\mathbf{h}_i$ in the representation space, the larger $\gamma$ value is desired, which can be characterized as follows

$$
\gamma_i^k = \frac{\exp(\mathbf{h}_i \diamond \mathbf{h}_k/\tau)}{\exp(\mathbf{h}_i \diamond \mathbf{h}_k/\tau) + \sum_{j \in \mathcal{B} \backslash (i, i+, k)} \exp(\mathbf{h}_i \diamond \mathbf{h}_j/\tau)} . \tag{3}
$$

That is, $\gamma_i^k$ approximates the relative importance of $\mathbf{h}_k$ to the anchor $\mathbf{h}_i$, among all $2N$-2 in-batch negatives. Despite the semantic equivalence between training examples except the given positive pairs are not available in our case, the above approximation of hard negatives is still valid as each training batch is randomly sampled with a much smaller size compared to that of the whole training data. Hence the presence of false negatives within each batch is negligible when the training data is large and diverse enough. We set the batch size to 8K in this paper, under which we observe monotonic increasing performance reported on the downstream tasks.

**Hard Positive**    We consider naturally occurring (text, function) as positive pairs, where the text is mined from the function docstring (Husain et al., 2019). The extracted text often summarizes the high-level semantics of the code. Therefore, contrastive learning with such bimodal data largely boosts the NL2Code semantic search performance in Section 4.2.2. Further, the extracted text of semantically equivalent code, no matter from the same or different programming languages, is often less diverse compared to the code themselves. Thereby, semantically similar codes can be implicitly grouped together through the same or very similar summary text. Our conjecture is validated by the large performance gain on both in-language and cross-language Code2Code search in Section 4.2.2.

---

[3]For example, masking a couple of tokens randomly from `tokenizer.convert_ids_to_tokens` can yield `tokenizer.convert_ids_to<mask><mask>` but random token replacement can result in `tokenizer.convert_jet_toboattokens`. Consequently, the code semantics are largely altered and representation learning via the self-attention mechanism can thereby deteriorate. See Appendix A.3 for more.

It is also easy to see that function names and input variable names often share a significant similarity, especially in terms of the lexical overlap with the summary text (see Appendix A.6 for statistics). We thereby form hard positives by removing both function signature and return statements.[4] We assess the effectiveness of such hard positive construction strategy in Section 4.2.2.

## 4 EXPERIMENTS

**Training Data and Model Architecture**   We train our models on The Stack dataset (Kocetkov et al., 2022) over nine languages - Python, Java, Javascript, Typescript, C#, C, Ruby, Go, and PHP. As aforementioned, we train three embedding models with size 130M (CODESAGE-SMALL), 356M (CODESAGE-BASE), and 1.3B (CODESAGE-LARGE) parameters. Please refer to Appendix A for training details at each stage and model hyper-parameters.

**Evaluation Protocol**   We assess the performance of our models over two main categories of downstream tasks, semantic search and classification. Our goal is to perform an evaluation of the encoder models for those practical scenarios where supervised fine-tuning data collection is costly. We thereby focus on *zero-shot* semantic search and only finetuning a linear classification layer on top of the frozen encoders for classification tasks (Peters et al., 2019; Chen et al., 2020; Wang et al., 2022). We report the fully finetuned classification results and finetuning hyper-parameters in Appendix B.3.

**Baselines**   We compare our models against four general-purpose code representation learning encoders and OpenAI-Embedding-Ada-002 (*For convenience, we refer to it as OpenAI-Ada-002. For comparative analysis with OpenAI-CPT-Code-001 and OpenAI-Text-Embedding-3, please see Tables 8 & 9 in Appendix, where OpenAI-Ada-002 attains significantly better or on par performance against the previous and latest models, respectively*). Both CodeBERT (Feng et al., 2020b) and GraphCodeBERT (Guo et al., 2021) are trained with standard MLM on six programming languages using CodeSearchNet (Husain et al., 2019)[5], while the replaced token detection objective (Clark et al., 2020) and data flow prediction objectives are adopted as auxiliary objectives, respectively. UnixCoder (Guo et al., 2022) is trained via three language modeling and two contrastive learning objectives using the same dataset. More recently, StarEncoder (Li et al., 2023) is trained with MLM and next sentence prediction (Devlin et al., 2019a) on 86 programming languages from The Stack (Kocetkov et al., 2022). We provide more details for each baseline model in Table 6 in Appendix. We also consider decoder-only baselines in Table 8 & 9 in Appendix B.

### 4.1 COMPARISON WITH THE BASELINES

We first compare CODESAGE against the aforementioned baselines on the following tasks.

**Code2Code** semantic search is the task of retrieving relevant code fragments given a code fragment as a *query*. In this work, we extend the Code2Code search evaluation set (Guo et al., 2022) created from CodeNet to six more languages - C, C#, Javascript, Typescript, GO, and PHP, for which we summarize the details in Appendix B.2. We report the in-language where query and candidate codes are in the same language, code2code search results in Table 1.

**NL2Code** semantic search is the task of using natural language as the query to retrieve the relevant code. We consider three benchmarks in Table 2, CoSQA (Huang et al., 2021), AdvTest (Lu et al., 2021), and CSN (Guo et al., 2021) . Detailed data statistics can be found in Appendix B.2.

**Classification** We consider three source code classification tasks. Code Defect detection is a benchmark in C from CodeXGLUE (Lu et al., 2021), with a binary label indicating whether a code is insecure and may attack software systems. Code Complexity prediction (Jeon et al., 2023) is a Java benchmark that requires predicting the algorithmic complexity among 7 labels. The RunTime error prediction (Bieber et al., 2023) benchmark has 29 possible labels with highly imbalanced distribution (see Table 10 in Appendix). For a more robust evaluation, we balance the dataset by aligning its total training examples of the "no_error" class with the cumulative count of the other 28 classes.

**Overall Performance Summary**   On Code2Code search, Table 1 shows that CODESAGE-SMALL (130M) persistently outperforms all the baseline models with known model size (*i.e.,* exclude

---

[4]Removal of function signature reduces the chance to learn shortcuts due to its similarity with the summary text. We remove the return statements to make a code look like a generic code snippet.

[5]The dataset includes 2.3M functions paired with natural language documents.

| Model | Python | Java | JS | TS | C# | C | Ruby | PHP | GO | Avg |
|---|---|---|---|---|---|---|---|---|---|---|
| CodeBERT | 14.40 | 7.62 | 5.47 | 6.05 | 3.66 | 5.53 | 13.55 | 10.28 | 6.27 | 8.09 |
| GraphCodeBERT | 19.23 | 10.78 | 7.38 | 8.65 | 5.54 | 8.48 | 19.69 | 15.67 | 9.65 | 11.68 |
| StarEncoder | 19.17 | 11.65 | 9.0 | 10.52 | 5.69 | 9.72 | 21.57 | 16.98 | 10.81 | 12.79 |
| UnixCoder | 30.77 | 16.45 | 21.32 | 21.95 | 6.19 | 15.62 | 32.33 | 31.93 | 13.94 | 21.17 |
| OpenAI-Ada-002 | 35.91 | 25.13 | 19.01 | 21.86 | 10.17 | 29.15 | 40.85 | 40.47 | 23.43 | 27.33 |
| CODESAGE-SMALL | 36.31 | 23.97 | 26.60 | 29.90 | 11.84 | 22.84 | 29.06 | 34.64 | 19.56 | 26.08 |
| CODESAGE-BASE | **47.52** | 22.84 | 28.70 | 31.95 | 13.37 | 30.99 | 44.86 | 51.13 | 25.15 | 32.95 |
| CODESAGE-LARGE | 46.70 | **33.13** | **37.16** | **41.18** | **16.81** | **32.89** | **54.12** | **52.13** | **32.48** | **38.51** |

Table 1: MAP score (%) of the zero-shot code search task. The language names mentioned in the top row indicate the languages queries and candidates are written in.

| | NL2Code | | | Classification | | |
|---|---|---|---|---|---|---|
| Model | CoSQA | AdvTest | CSN | Defect | Complexity | RunTime |
| CodeBERT | 0.24 | 0.06 | 0.10 | $51.82_{0.38}$ | $35.60_{1.96}$ | $6.2_{0.02}$ |
| GraphCodeBERT | 16.20 | 5.58 | 11.26 | $55.26_{0.28}$ | $55.54_{1.98}$ | $10.63_{0.10}$ |
| StarEncoder | 10.78 | 0.93 | 2.69 | $53.2_{0.11}$ | $50.63_{3.33}$ | $8.91_{0.05}$ |
| UnixCoder | 42.11 | 27.32 | 46.39 | $60.28_{0.04}$ | $76.45_{1.10}$ | $20.87_{0.43}$ |
| OpenAI-Ada-002 | 44.23 | 38.08 | 71.24 | $\mathbf{62.56}_{0.11}$ | $79.82_{0.50}$ | $20.84_{0.36}$ |
| CODESAGE-SMALL | 49.92 | 41.28 | 63.86 | $57.52_{0.21}$ | $79.76_{0.50}$ | $\mathbf{25.05}_{1.04}$ |
| CODESAGE-BASE | 48.50 | 49.08 | 68.72 | $57.74_{0.09}$ | $85.32_{1.72}$ | $24.70_{0.40}$ |
| CODESAGE-LARGE | 47.53 | **52.67** | 71.24 | $58.95_{0.13}$ | $\mathbf{90.32}_{2.10}$ | $24.42_{0.28}$ |

Table 2: **Left.** MRR score (%) of NL2Code search in zero-shot setting. For CSN, we report the average performance over six languages (see Table 9 in Appendix for the detailed results). **Right.** F1 (macro) score of the source code classification tasks attained by only finetuning the classification head. We finetuned each model using three seeds and reported the mean and standard deviation (in subscript). The fully finetuned results can be found in Appendix B.3.

OpenAI-Ada-002) on every language, with 23.19% relative (4.91% absolute) improvement on the average performance when comparing with UnixCoder. With the increased model size, CODESAGE-BASE and CODESAGE-LARGE outperform the best baseline model, *i.e.,* OpenAI-Ada-002 (model size unknown), with 20.56% relative (5.62% absolute) and 40.91% relative (11.18% absolute) improvement on the average performance, respectively.

As shown in Table 2, CODESAGE-SMALL achieves 18.54% to 51.1% relative (7.81% to 13.96% absolute) improvement over UnixCoder on NL2Code search. Compared to OpenAI-Ada-002, CODESAGE-SMALL attains a 12.86% relative (5.69% absolute) improvement on CosQA and an 8.4% relative (3.12% absolute) improvement on AdvTest. On the other hand, OpenAI-Ada-002 attains the same average performance as CODESAGE-LARGE on CSN. However, we want to highlight the performance gain attained by CODESAGE on AdvTest which contains normalized Python functions (from CSN) with function and variable names replaced by dummy variables (see Figure 9 in Appendix). AdvTest constructed in this way better assesses the generalization performance as the model needs to understand what the obfuscated code does so as to identify the correct target code for a given natural language query.

Compared to both UnixCoder and OpenAI-Ada-002, CODESAGE persistently performs better on code complexity and runtime error prediction with large margins in Table 2. We also notice that CODESAGE underperforms both models on code defect detection, whilst attaining better performance when we finetuning the full models in Table 12 in Appendix.

## 4.2 ABLATION STUDY

### 4.2.1 MASKING STRATEGY

**80-10-10 vs. Full Mask** Given an input sequence, standard MLM (Devlin et al., 2019b) first randomly samples a subset of its tokens, of which 80% are replaced by a special token "[MASK]", 10% are left unchanged, and the other 10% are replaced by random tokens from the vocabulary. We revisit

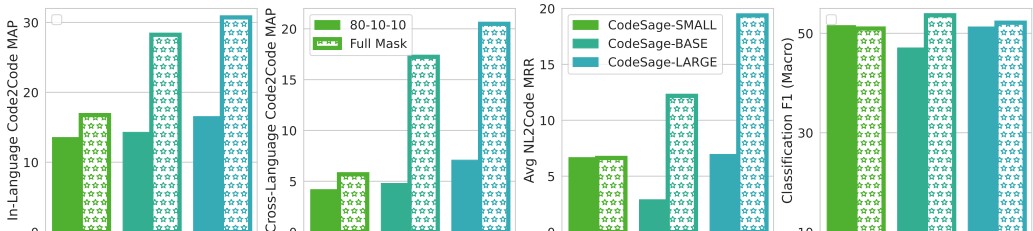

```
1  def binary_search(arr, low, high, x):
2      '''Returns index of x in arr if
3      present, else -1.'''
4      if high >= low:
5          mid = (high + low) // 2
6          if arr[mid] == x:
7              return mid
8          elif arr[mid] > x:
9              return binary_search(
10                 arr, low, mid - 1, x)
11         else:
12             return binary_search(
13                 arr, mid + 1, high, x)
14     else:
15         return -1
```

```
1
2  def chem_search(arr, low<MASK> high, x):
3      '''Returns <MASK> of x in arr if
4      present, else -<MASK> getConfig
5      if sal >= lownone
6          mid = (high <MASK> low) <MASK> 2
7      if arr[mid] == x:
8          return mid
9      elif <MASK>[mid] > x:
10             return 的所有_<MASK>(
11                 arr, low, mid - 1, <MASK>)
12     else:
13         return instead_search(
14             arr, mid + 1, highsystemd x)
15     else:
16         return -1
```

(a) Sample code (left) and its corrupted version following the 80-10-10 rule (right).

(b) With a fixed masking rate of 15%, we assess the effectiveness of applying "Full Mask", *i.e.,* replacing the sampled tokens with the [MASK] token only, and the 80-10-10 corruption strategy on downstream tasks.

Figure 2: 80-10-10 vs. "Full Mask".

| Model | CODESAGE-SMALL | | | | CODESAGE-BASE | | | | CODESAGE-LARGE | | | |
|---|---|---|---|---|---|---|---|---|---|---|---|---|
| | R | D | S | P | R | D | S | P | R | D | S | P |
| NL2Code | 6.6 | 19.9 | 22.7 | 25.8 | 12.2 | 22.5 | 22.0 | 23.3 | 19.4 | 23.3 | 29.4 | 30.5 |
| Code2Code (In) | 16.8 | 14.6 | 17.9 | 19.7 | 28.2 | 23.7 | 25.3 | 29.2 | 30.7 | 28.2 | 30.2 | 33.9 |
| Code2Code (Cross) | 5.7 | 6.7 | 8.8 | 9.6 | 17.2 | 14.1 | 14.6 | 19.7 | 20.5 | 18.0 | 19.0 | 24.6 |
| Classification | 51.2 | 53.9 | 53.5 | 53.4 | 53.8 | 55.6 | 54.8 | 55.4 | 52.0 | 55.6 | 57.2 | 56.5 |

Table 3: We explore two options to leverage DOBF (D) and random masking (R) to complement each other. (1) Sequential (S): training the model with random masking first, then DOBF. (2) Parallel (P): randomly picking either DOBF or random masking for a training example – our strategy.

the effectiveness of such convention, originally proposed for text, for code in Figure 2. Surprisingly, compared to simply replacing all selected tokens with the [MASK] token, *i.e.,* "Full Mask", the 80-10-10 masking scheme causes a large performance drop across different downstream tasks, as shown in Figure 2b. A similar finding has been reported in Gao et al. (2022) for text. However, the degradation is more severe for source code. As Figure 2a indicates, when replacing with random tokens, both the semantics and structure of the masked code can be largely disrupted, which together with the presence of "[MASK]" tokens makes the learning too challenging (see Appendix A.3 for more discussions). We hypothesize that excessive corruption may also account for the modest enhancement observed in downstream tasks when scaling up the size of a model trained with 80-10-10 in Figure 2b. It would be intriguing to explore whether this scaling trend would experience a sudden expansion with a further increase in model size and training data, potentially identifying a phase transition point, provided that the computational resources permit such an investigation.

**Deobfuscation & Random Masking Complement Each Other** We investigate DOBF and the random masking based MLM with "Full Mask" in Figure 3. DOBF persistently outperforms random masking on classification, which validates our motivation that the model is promoted to better capture (understand) the code structure so as to predict the identifier names. DOBF also performs better on NL2Code search than random masking. A potential reason could be natural language in comments and docstrings often carry rich semantics of code while both being excluded from masking in DOBF; hence when training the model to predict the identifier names, it will look at and correlate with the natural language and lead to better contextualized representations between natural

language and programming language. On the other hand, the random masking strategy (with "Full Mask") outperforms DOBF on both in-language and cross-language Code2Code search tasks. As examined in Appendix A.3, a large portion of tokens in code snippets are not identifiers. Therefore, the random masking strategy allows the model to learn beyond identifiers and enrich the semantics encoded in representations. In summary, Table 3 validates our strategy of jointly optimizing DOBF and random masking so as to leverage their strengths to complement each other.

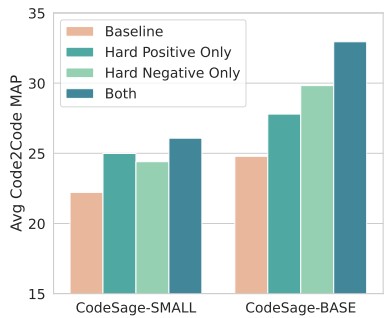
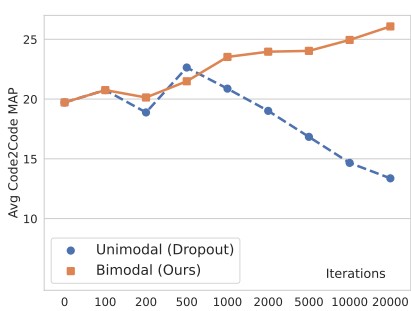

(a) Effectiveness of hard negatives and hard positives.     (b) Unimodal vs. bimodal contrastive learning.

Figure 3: **(a)** Hard negative and hard positive can independently boost performance over the baseline where neither is applied. Further improvement is attained when leveraging them simultaneously. **(b)** Unimodal contrastive learning with positives obtained via dropout requires longer training and hence cannot leverage vast amounts of training data to further enhance the representations.

### 4.2.2 ON EFFECTIVENESS OF CONTRASTIVE LEARNING

**Hard Positive and Hard Negative Effectively Boost Performance**   We first demonstrate the effectiveness of the hard positive and hard negative construction strategy in Figure 3a. As it shows, both hard positive and hard negative can independently improve the performance by a large margin, while the combination of them persistently yields better performance across different model sizes. We also observe that a large model size (*i.e.,* CODESAGE-BASE) benefits more from the proposed hard negative construction strategy. This observation is unsurprising since larger models possess more capacity to leverage more challenging and effective learning objectives.

**Unimodal vs. Bimodal Contrastive Learning**   In Figure 3b, we compare our bimodal contrastive learning approach against the *Dropout*-based unimodal contrastive learning where a positive pair is obtained by leveraging different dropout masks of the transformer in two forwarding passes of the same sequence (Gao et al., 2021; Guo et al., 2022). For a fair comparison, hard negative optimization is applied to both approaches. We can see that the *dropout*-based unimodal contrastive learning suffers from supporting a long training process and hence cannot effectively utilize a large amount of pretraining data to further improve the representations. A similar finding has been reported by (Zhou et al., 2022). Indeed, both Gao et al. (2021) nor Guo et al. (2022) – demonstrate dropout as effective augmentation for text and code respectively, only use a few million training examples that can be covered by the amount of training data in the first 500 iterations (with batch size 8K) in Figure 3b where the *dropout*-based contrastive learning shows improvement over the baseline.

**Larger Improvement on Cross-Lingual Search**   To gain a deeper understanding of the performance improvement achieved through contrastive learning during Stage II of pretraining, we delve into the analysis of semantic search performance. As Figure 4a shows, contrastive learning persistently boosts the search performance with comparatively larger improvement on the cross-lingual scenarios, encompassing both NL2Code and cross-language Code2Code search. We posit that the text extracted from docstring helps group semantically equivalent code together as the text often summarizes the high-level semantics of code and hence are likely less diverse than the code themselves. In particular, those parallel examples from different programming languages can share very similar or even the same summary. For NL2Code, the larger improvement can be credited to its alignment with the bimodal contrastive learning objective using *(text, code)* as positives. Such bimodal objective also brings NL and PL closer in Figure 4b. Compared to the model trained at Stage-I only, contrastive learning pulls together NL and PL such that the relative similarity gap between parallel NL2Code pairs and cross-language Code2Code parallel examples largely decreased.

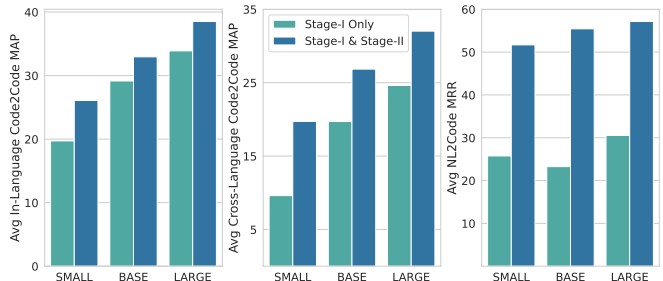 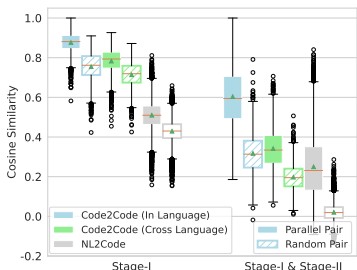

(a) The performance of CODESAGE in semantic search, comparing results between searches within the same language and across different languages, while varying model sizes and training approaches.

(b) Cosine similarity between parallel examples vs. randomly sampled pairs using CODESAGE representations.

Figure 4: Examining the effectiveness of contrastive learning (Stage-II) by comparing CODESAGE against those trained with the token-level denoising objective only (Stage-I). **(a)** Compared to the in-language Code2Code search, contrastive learning persistently leads to a larger performance boost for cross-lingual search, including both NL2Code and cross-language Code2Code search. **(b)** Contrastive learning leads to more dispersed representation space with improved discrimination, as indicated by the corresponding enlarged similarity gap between parallel and randomly sampled pairs, while simultaneously bridging the relative similarity gap between NL2Code and Code2Code pairs.

## 4.3 ON OBJECTIVE AND DOWNSTREAM PERFORMANCE SCALING WITH MODEL SIZE

In Figure 5, we study how the downstream task performance scales with the model size when pretrained with different schemes, *i.e.,* token-level objective only (Stage-I), contrastive learning only (Stage-II), and our proposed two-stage framework with Stage-I followed by Stage-II. We use *zero-shot* multilingual in-language code search performance (averaged over nine languages) for this exploration. We can see that models pretrained from scratch with contrastive learning alone do not scale with the increased model size. Neelakantan et al. (2022) report a similar finding that the contrastive objective on its own is not sufficient to learn useful representations. When training from scratch with contrastive learning only, we find the training loss often converges at a large value, indicating the model cannot well discriminate each positive pair from the other in-batch negatives. In other words, leveraging the token-level denoising objective to provide a good embedding foundation is essential for contrastive learning to be effective and further enhance the sequence-level presentations.

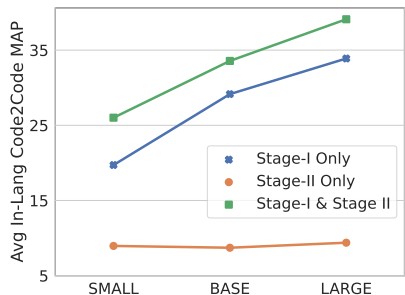

Figure 5: On the downstream task performance scaling with pretrained model size under different training schemes.

## 5 CONCLUSION

In this study, we unveiled CODESAGE, a cutting-edge encoder representation learning model for source code. We trained CODESAGE using an extensive dataset comprising 237 million code files and 75 million bimodal code and natural language pairs across nine languages. Our findings reveal that our model outperforms its predecessors significantly in tasks related to code search and code classification. We also delve into the essential factors contributing to enhanced code representation learning across various model sizes.

We hope our work will serve as an inspiration for future works not only in code representation learning by effectively utilizing publicly accessible extensive corpora for source code, but also in the broader field of `universal` model training. This includes integrating language generation and embedding within a single model, as seen in the works of (Jain et al., 2023; Muennighoff et al., 2024). Additionally, we aim to encourage advancements in cross-domain representation learning, such as unifying text and code embedding within a single model.

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

## A    Data, Model, and Hyper-parameters Details

### A.1    Pretraining Data

**Masked Language Modeling (MLM) and Identifier Deobsfucation (DOBF)**    For both MLM and DOBF, we use the Stack dataset (Kocetkov et al., 2022). We set the maximum sequence length to 1024 with concatenation and block attention.

**Contrastive Learning (CL)**    In CL, we focus on bimodal data, *i.e.,* code and natural language pairs, denoted as *(text, function)*. Text is extracted as the first sentence from the docstring of a function (Husain et al., 2019). For better interpretation, we refer to such text as "summary" in this section as it often summarizes the high-level semantics of a function. We filter or modify summaries based on the following practices.

1. Filter summary if it is not in English.
2. Filter summary if the number of tokens in a summary is <3 or >256.
3. Remove URLs, HTML tags, and doctags from the summaries.
4. Fix bad Unicode text in the summaries.
5. Filter functions with no more than one line of code in the function body.

We summarize the statistics of our pretraining data at each stage in Figure 1 and Table 4.

| Language | Total files | #Functions | #Func. w/ docstring | #Func. w/ summary |
|---|---|---|---|---|
| Python | 24,214,270 | 67,264,716 | 24,321,126 | 18,146,327 |
| Java | 42,429,211 | 84,828,833 | 17,613,636 | 13,118,303 |
| Javascript | 40,112,121 | 35,469,803 | 7,450,153 | 4,796,101 |
| C# | 21,702,269 | 37,284,300 | 9,325,665 | 7,350,191 |
| C | 21,383,832 | 16,253,435 | 4,392,973 | 2,958,699 |
| Ruby | 7,205,146 | 5,475,047 | 1,217,764 | 1,049,356 |
| GO | 11,653,185 | 31,067,259 | 11,247,051 | 9,739,861 |
| PHP | 34,851,418 | 42,373,766 | 22,191,329 | 13,416,574 |
| Typescript | 19,589,267 | 16,612,988 | 2,637,245 | 1,863,436 |
| Total | 237,961,548 | 367,905,026 | 105,760,862 | 75,389,347 |

Table 4: Statistics of the data used in pre-training via Masked Language Modeling (MLM) and Identifier Deobsfucation (DOBF), followed by contrastive learning (CL). The data is collected from The Stack (Kocetkov et al., 2022).

### A.2    Model and Training Hyper-parameters

We pretrain three sizes of model architecture which we refer to as CodeSage-small, CodeSage-base, and CodeSage-large. We summarize the model hyper-parameters in Table 5.

### A.3    On Token Distribution and Stage-I Pretraining Objective

In our preliminary study, we perform data analysis where we examine the ratio of natural language (NL) and programming language (PL) tokens. In a source code, tokens are broadly categorized into five groups: (1) identifiers, (2) keywords, (3) operators, (4) delimiters, and (5) literals. We tag the *String literals* (*i.e.,* docstring, comments) as NL tokens, while all other tokens are considered PL tokens. We use *tree-sitter* to parse source code and extract the five categories of code tokens. Then we tokenize them using Starcoder tokenizer (Li et al., 2023). From Stack-Python corpora, we compute the following statistics using Starcoder tokenized tokens.

1. *Approximate PL Tokens*: 57.8% of tokens belong to {identifiers, keywords, delimiters, operators}. Among them, 53.8% tokens belong to identifiers and 46.2% are other tokens.
2. *Approximate NL Tokens*: 42.2% of tokens Literals {Boolean, Numeric, String}. Among them, 92.9% tokens belong to String literals and 7.1% tokens belong to others.

As we can tell from the above numbers, the approximate NL tokens account for roughly 40% of the overall tokens for a particular programming language. Therefore, when replacing masked to-

|  | CODESAGE-SMALL | CODESAGE-BASE | CODESAGE-LARGE |
|---|---|---|---|
| #layers | 6 | 24 | 24 |
| #heads | 8 | 8 | 16 |
| Model dim | 1024 | 1024 | 2048 |
| Vocab size | 49,152 | 49,152 | 49,152 |
| Max sequence length | 1024 | 1024 | 1024 |
| Total parameters | 130M | 356M | 1.3B |
| Stage1: Masked Language Modeling | | | |
| Dropout | 0.1 | 0.1 | 0.1 |
| Max steps | 250,000 | 250,000 | 250,000 |
| Warmup steps | 5000 | 5000 | 5000 |
| Batch size | 2048 | 2048 | 2048 |
| Base learning rate | 3e-4 | 3e-4 | 3e-4 |
| Stage2: Contrastive Learning | | | |
| Dropout | 0.1 | 0.1 | 0.1 |
| Max steps | 20,000 | 20,000 | 20,000 |
| Warmup steps | 500 | 500 | 500 |
| Batch size | 8192 | 8192 | 8192 |
| Bae learning rate | 5e-06 | 5e-06 | 5e-06 |

Table 5: Model architecture and pre-training related hyper-parameters.

kens with a random token could result in replacing a PL token with an NL token, and vice versa. However, there are often no clear boundaries between PL and NL tokens in many scenarios, as PL tokens, *e.g.,* those identifier-related tokens, are often expected to carry clear semantics so that the code snippets are interpretable by humans. Therefore, given masked input tokens following the 80-10-10 convention, it can be a non-trivial task for the model to decide which tokens are from corruption. This together with the structure nature of PL makes it possible for those random tokens to largely disrupt both the semantics and structure of code and make the representation learning too challenging to be effective.

Take the example in Figure 6 (right) for illustration, the function name "binary_search" is being corrupted with random tokens at all three places it appears, which has the potential to alter the semantic meaning of the code. Although we may expect the model to still be able to correctly recover "binary_search" from the corrupted code, it is a challenging task given (1) the syntax of the code has been disrupted by another random token "getConfig"; (2) the presence of $<$ MASK $>$ tokens; and (3) the bidirectional self-attention mechanism can drive the model to leverage those random tokens to form prediction of the masked tokens.

```
1  def binary_search(arr, low, high, x):
2      '''Returns index of x in arr if
3      present, else -1.'''
4      if high >= low:
5          mid = (high + low) // 2
6          if arr[mid] == x:
7              return mid
8          elif arr[mid] > x:
9              return binary_search(
10                 arr, low, mid - 1, x)
11         else:
12             return binary_search(
13                 arr, mid + 1, high, x)
14     else:
15         return -1
```

```
1  def chem_search(arr, low<MASK> high, x):
2      '''Returns <MASK> of x in arr if
3      present, else -<MASK> getConfig
4      if sal >= lownone
5          mid = (high <MASK> low) <MASK> 2
6          if arr[mid] == x:
7              return mid
8          elif <MASK>[mid] > x:
9              return 的所有_<MASK>(
10                 arr, low, mid - 1, <MASK>)
11         else:
12             return instead_search(
13                 arr, mid + 1, highsystemd x)
14     else:
15         return -1
```

Figure 6: A code snippet(on the left) and its corresponding masked version, were created using the 80-10-10 practice with a 15% masking rate (on the right).

## A.4 MASKING RATE

With "Full Mask", *i.e.,* MLM without the 80-10-10 corruption strategy, we investigate the optimal masking rate in Figure 7. We consider three constant masking rates, 7.5%, 15%, and 30%, as well

as a dynamic masking strategy with the masking rate being randomly selected from the range [10%, 50%] for each training example. We find $15\%$ remains the optimal masking rate among the four variants we investigate.

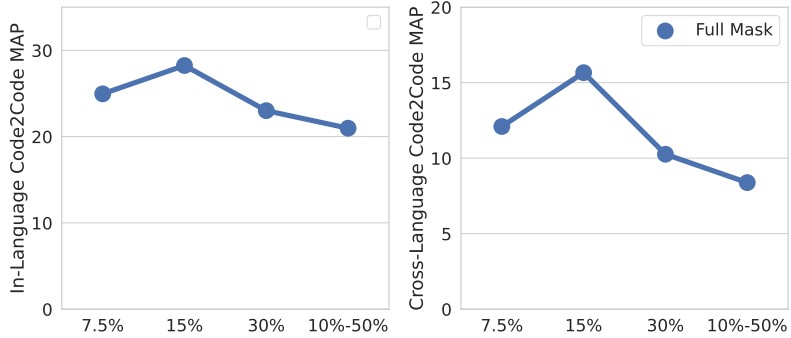

Figure 7: Maksing rate and zero-shot Code2Code search performance investigated on CODESAGE-BASE. We consider three constant masking rates, 7.5%, 15%, and 30%, as well as a dynamic masking strategy with the masking rate being randomly selected from the range [10%, 50%] for each training example.

## A.5 IDENTIFIER OBFUSCATION

In this research, we employ an identifier deobfuscation (DOBF) objective to train bidirectional encoder representation models. While our inspiration for this approach comes from the DOBF method introduced by anne Lachaux et al. (2021), our adoption strategy differs from theirs. In their work, anne Lachaux et al. (2021) trained a sequence-to-sequence language model to reconstruct the original code from an obfuscated version where class, function, and variable names were replaced with special tokens. In contrast, our approach applies this technique to encoder-only models. This adaptation involves a non-trivial effort to establish a 1-1 mapping between mask tokens and identifier tokens (will be masked and encoders will predict them) due to disparities in code tokenization (*i.e.,* using *tree-sitter*) and model-specific tokenization (*i.e.,* utilizing a *sentencepiece* tokenizer). To illustrate, let's consider the tokenization process. Tree-sitter tokenizes *"def function_name():"* as $\{def, function\_name, (,), :\}$, whereas a model-specific tokenizer might tokenize it as $\{def, function, \_, name(,), :\}$. Consequently, we encounter a challenge to construct the mapping from masked tokens to prediction tokens: $\{[mask], [mask], [mask]\} \rightarrow \{function, \_, name\}$, by skipping "(" token that is part of the identifier token "name". To perform obfuscation and construct the *mask map*, we developed an obfuscation (OBF) tool.

**OBF Tool**   We developed a tool that takes an entire source code file or a function as input and outputs an identifier obfuscated code along with a token map. We provide an example in Figure 8. We used *tree-sitter* to parse a code snippet and extract all identifiers and their ancestor node types. Based on the node types, we identify class names, function names, function arguments, and function calls. Then we replace them with special tokens ($c_i, f_i, v_i$ for class names, function names, and variable names, respectively). Then we include the special tokens into the model tokenizer (*i.e.,* Starcoder tokenizer) and tokenize the obfuscated code such that special tokens are retained in the output tokens. Finally, we use the model tokenizer to tokenize the identifiers individually and replace the special tokens ($c_i, f_i, v_i$) with the identifier tokens.

## A.6 ON LEXICAL OVERLAP AND HARD POSITIVE DESIGN

As detailed in Appendix A.1, we extract the first sentence from the function docstring as the summary text. We then examine the lexical overlap between docstring (summary) and function signature versus that between docstring (summary) and function body. In Stack-Python corpora, we found -

1. 22.3% of tokens in function signature and 23.1% of tokens in function body overlap with docstring tokens.
2. 12.3% of tokens in function signature and 11.6% of tokens in function body overlap with summary tokens.

```python
class Node:
    def __init__(self, v):
        self.data = v
        self.left = None
        self.right = None

# Function to print postorder traversal
def printPostorder(node):
    if node == None:
        return

    # First recur on the left subtree
    printPostorder(node.left)

    # Then recur on the right subtree
    printPostorder(node.right)

    # Now deal with the node
    print(node.data, end=' ')
```

```python
class c_0:
    def f_0(v_0, v_1):
        v_0.v_2 = v_1
        v_0.v_3 = None
        v_0.v_4 = None

# Function to print postorder traversal
def f_1(v_5):
    if v_5 == None:
        return

    # First recur on the left subtree
    f_1(v_5.v_3)

    # Then recur on the right subtree
    f_1(v_5.v_4)

    # Now deal with the node
    print(v_5.v_2, end=' ')
```

Figure 8: An example of a Python code (at the left) and its corresponding obfuscated version (at the right) generated by our developed obfuscation tool. The class names, function names, and variables are replaced by special tokens. Given the code on the left, our developed OBF tool produces the obfuscated code and the identifier map: $\{c_0, v_0, v_1, v_2, v_3, v_4, v_5, f_0, f_1\} \rightarrow \{Node, self, v, data, left, right, node, \_\_init\_\_, printPostorder\}$.

This validates our intuition that the docstring or summary of a function often has a large lexical overlap with the function signature. Thereby, when contrasting docstring or summary with the entire function, the model tends to learn shortcut by leveraging such overlap, and hence fail to capture the semantic equivalence concept in the representations. Consequently, poor generalization is attained.

## B EVALUATION OF DOWNSTREAM TASKS

### B.1 BASELINE MODELS

We summarize the baseline model size and output representation dimension in Table 6.

| Model | Model Size | Embedding Dimension | Max Sequence Length | Training Data Source |
|---|---|---|---|---|
| CodeBERT | 125M | 768 | 512 | CodeSearchNet |
| GraphCodeBERT | 125M | 768 | 512 | CodeSearchNet |
| StarEncoder | 125M | 768 | 1024 | The Stack |
| UnixCoder | 125M | 768 | 1024 | CodeSearchNet |
| OpenAI-Embedding-Ada-002 | Unknown | 1536 | 8191 | Unknown |

Table 6: Model size and dimension of the embeddings. The GitHub Code dataset is available at (https://huggingface.co/datasets/codeparrot/github-code).

### B.2 CODE SEARCH

We summarize the data statistics of NL2Code and Code2Code benchmarks in Table 7. Below, we provide more context on each dataset.

**Code2Code** search is the task of retrieving relevant code fragments given a code fragment as a *query*. In this work, we extend the code2code search dataset (Guo et al., 2022) created from CodeNet to six more languages - C, C#, Javascript, Typescript, GO, and PHP. The original dataset includes 2 to 10 solutions for each problem in Java, Python, and Ruby. At first, we collect the problem identifiers and aggregate solutions in those six languages from CodeNet. Also, CodeNet provides cluster identifiers for each solution where solutions within a cluster are near duplicates of each other. We collect one

| Code2Code Semantic Search Data Statistics in Each Language | | | | | | | | |
|---|---|---|---|---|---|---|---|---|
| | Python | Java | JS | TS | C# | C | Ruby | PHP | GO |
| Num Queries | 15,594 | 23,530 | 6,866 | 3,385 | 11,952 | 11,260 | 11,744 | 6,782 | 9,720 |
| Num Candidates | 15,594 | 23,530 | 6,866 | 3,385 | 11,952 | 11,260 | 11,744 | 6,782 | 9,720 |

| NL2Code Semantic Search Data Statistics in Benchmark and Language | | | | | | | | |
|---|---|---|---|---|---|---|---|---|
| | CoSQA | AdvTest | CSN | | | | | |
| | Python | Python | Python | Java | JS | PhP | Go | Ruby |
| Num Queries | 500 | 19,120 | 14,918 | 10,955 | 3,291 | 14,014 | 8,122 | 1,261 |
| Num Candidates | 6,268 | 19,120 | 43,827 | 40,347 | 13,981 | 52,660 | 28,120 | 4,360 |

Table 7: Evaluation data statistics of both NL2Code and Code2Code search.

| Model | Python | Java | JS | TS | C# | C | Ruby | PHP | GO | Avg |
|---|---|---|---|---|---|---|---|---|---|---|
| CodeGen2.5(7B) | 16.45 | 10.18 | 7 | 8.46 | 4.24 | 7.99 | 17.28 | 15.56 | 9.38 | 10.73 |
| Starcoder(15.5B) | 7.09 | 3.91 | 3.19 | 4.36 | 1.72 | 2.35 | 6.82 | 6.06 | 3.26 | 4.31 |
| CodeT5+(16B) Encoder | 18.24 | 9.85 | 5.84 | 6.86 | 4.19 | 8.16 | 16.47 | 13.88 | 8.01 | 10.17 |
| CodeBERT | 14.40 | 7.62 | 5.47 | 6.05 | 3.66 | 5.53 | 13.55 | 10.28 | 6.27 | 8.09 |
| GraphCodeBERT | 19.23 | 10.78 | 7.38 | 8.65 | 5.54 | 8.48 | 19.69 | 15.67 | 9.65 | 11.68 |
| StarEncoder | 19.17 | 11.65 | 9.0 | 10.52 | 5.69 | 9.72 | 21.57 | 16.98 | 10.81 | 12.79 |
| UnixCoder | 30.77 | 16.45 | 21.32 | 21.95 | 6.19 | 15.62 | 32.33 | 31.93 | 13.94 | 21.17 |
| OpenAI-CPT-Code-001 | 21.92 | 8.90 | 4.90 | 5.70 | 3.15 | 11.58 | 26.25 | 16.60 | 9.40 | 12.04 |
| OpenAI–Ada-002 | 35.91 | 25.13 | 19.01 | 21.86 | 10.17 | 29.15 | 40.85 | 40.47 | 23.43 | 27.33 |
| OpenAI-Text-3-Small | 25.18 | 12.61 | 8.00 | 9.44 | 5.46 | 15.86 | 30.70 | 23.33 | 11.2 | 15.75 |
| OpenAI-Text-3-Large | 40.57 | 25.33 | 20.09 | 22.00 | 11.84 | 31.9 | 42.54 | 41.84 | 21.75 | 28.65 |
| CODESAGE-SMALL | 36.31 | 23.97 | 26.60 | 29.90 | 11.84 | 22.84 | 29.06 | 34.64 | 19.56 | 26.08 |
| CODESAGE-BASE | **47.52** | 22.84 | 28.70 | 31.95 | 13.37 | 30.99 | 44.86 | 51.13 | 25.15 | 32.95 |
| CODESAGE-LARGE | 46.70 | **33.13** | **37.16** | **41.18** | **16.81** | **32.89** | **54.12** | **52.13** | **32.48** | **38.51** |

Table 8: MAP score (%) of the zero-shot code search task. The language names mentioned in the top row indicate the languages queries and candidates are written in.

solution from each cluster and randomly pick 2 to 10 solutions per problem. We summarize in-language (query and candidates are in the same language) code2code search results in Table 1.

**NL2Code** search refers to the task of using natural language as the query to retrieve the relevant code. We consider three benchmarks in this paper. CoSQA where the NL queries (in NL) are from the search logs of the Bing search engine and the candidate functions are from CodeSearch-Net (in Python). Total queries 500 and number of candidates 6,268. CSN is constructed from the CodeSearchNet dataset of six programming languages, including Python, Java, JavaScript, PHP, Go, and Ruby. AdvTest which normalizes Python functions (from CSN) and variable names to test the understanding and generalization capabilities of models (an example is shown in Figure 9).

**Additional Baselines** are considered in Tables 8 and 9. We constantly find decoder-only models yield poor performance on semantic search. Finetuning or prompt engineering may help improve the performance of decoder-only models, which we leave as future work.

We also compare against OpenAI-CPT-Code-001, specifically choosing the code-search-babbage-code-001 model, along with OpenAI-Text-3-Large and OpenAI-Text-3-Small. For the OpenAI-Text-3 models, we utilized their default embedding sizes of 3072 and 1536 for the large and small models, respectively. When submitting our paper in September 2023, based on OpenAI's recommendation, we evaluated against the OpenAI-Ada-002 model. Our findings, presented in Tables 8 & 9, indeed demonstrate that OpenAI-Ada-002 surpasses OpenAI-CPT-Code-001 in both Code2Code and NL2Code search tasks. Furthermore, while OpenAI-Text-3-Large shows superior performance over OpenAI-Ada-002, OpenAI-Text-3-Small falls short. Nevertheless, our model, CodeSage, consistently outperforms OpenAI-Text-3-Large in Code2Code search tasks and shows slightly inferior results in NL2Code search.

| Model | CoSQA Python | AdvTest Python | CSN Python | Java | JS | PhP | Go | Ruby |
|---|---|---|---|---|---|---|---|---|
| CodeGen2.5 (7B) | 0.02 | 0.01 | 0.06 | 0.02 | 0.05 | 0.18 | 6.03 | 2.04 |
| Starcoder (15.5B) | 0.02 | 0.06 | 0.03 | 0.01 | 0.05 | 0.59 | 0.06 | 0.05 |
| CodeT5+ (16B) Encoder | 22.96 | 20.36 | 19.93 | 14.05 | 12.26 | 26.08 | 20.37 | 13.05 |
| CodeBERT | 0.24 | 0.06 | 0.05 | 0.03 | 0.04 | 0.02 | 0.14 | 0.34 |
| GraphCodeBERT | 16.20 | 5.58 | 10.37 | 8.59 | 7.29 | 8.07 | 12.47 | 20.79 |
| StarEncoder | 10.78 | 0.93 | 2.81 | 2.51 | 1.87 | 0.74 | 2.65 | 5.54 |
| UnixCoder | 42.11 | 27.32 | 42.17 | 43.92 | 40.46 | 35.21 | 61.39 | 55.22 |
| OpenAI-CPT-Code-001 | 52.20 | 36.03 | 63.13 | 67.85 | 62.30 | 57.47 | 85.22 | 69.28 |
| OpenAI-Text-Ada-002 | 44.23 | 38.08 | 68.02 | 71.49 | 67.50 | 60.62 | 85.63 | 74.20 |
| OpenAI-Text-3-Small | 52.48 | 34.10 | 62.62 | 65.87 | 60.28 | 54.85 | 81.96 | 67.57 |
| OpenAI-Text-3-Large | **55.21** | 46.83 | **70.81** | **72.89** | 68.12 | 59.58 | **87.6** | **75.22** |
| CODESAGE-SMALL | 49.92 | 41.28 | 64.38 | 63.19 | 60.01 | 54.71 | 77.66 | 63.20 |
| CODESAGE-BASE | 48.50 | 49.08 | 67.99 | 68.02 | 66.95 | 58.15 | 83.21 | 68.00 |
| CODESAGE-LARGE | 47.53 | **52.67** | 70.77 | 70.21 | **69.50** | **61.33** | 83.71 | 71.92 |

Table 9: MRR score (%) of NL2Code search in zero-shot setting.

**NL query**: Try loading the given cache file.

```python
# Original Python function
def from_file(cls, file, *args, **kwargs):
    try:
        cache = shelve.open(file)
        return cls(file, cache, *args, **kwargs)
    except OSError as e:
        logger.debug("Loading {0} failed".format(file))
        raise e
```

```python
# AdvTest Python function
def Func(arg_0, arg_1, *arg_2, **arg_3):
    try:
        arg_4 = shelve.open(arg_1)
        return arg_0(arg_1, arg_4, *arg_2, **arg_3)
    except OSError as e:
        logger.debug("Loading {0} failed".format(arg_1))
        raise e
```

Figure 9: An example of natural language query and the associated ground truth function from the AdvTest dataset. The function names and variables in the original function (at the top) are replaced by special tokens (at the bottom) to obfuscate the code.

| Target Class | Train # | Valid # | Test # | Target Class | Train # | Valid # | Test # |
|---|---|---|---|---|---|---|---|
| No error | 1,20,503 | 13,049 | 13,745 | ImportError | 259 | 37 | 22 |
| ZeroDivisionError | 25,087 | 3,087 | 2,828 | TabError | 74 | 4 | 3 |
| OSError | 21540 | 2,427 | 2,422 | re.error | 62 | 6 | 11 |
| UnboundLocalError | 21,414 | 2,641 | 2,603 | AttributeError | 47 | 4 | 8 |
| decimal | 10,026 | 509 | 1,674 | StopIteration | 24 | 5 | 3 |
| ValueError | 8,585 | 991 | 833 | OverflowError | 19 | 2 | 2 |
| AssertionError | 7,816 | 1,072 | 691 | Timeout | 18 | 8 | 2 |
| FileNotFoundError | 7,676 | 727 | 797 | IndexError | 10 | 0 | 12 |
| IndentationError | 7,645 | 285 | 841 | ModuleNotFoundError | 8 | 7 | 1 |
| KeyError | 7,505 | 965 | 733 | RecursionError | 5 | 0 | 0 |
| NameError | 1,876 | 186 | 110 | EOFError | 3 | 0 | 0 |
| numpy.AxisError | 437 | 47 | 125 | SyntaxError | 3 | 0 | 1 |
| MathDomainError | 362 | 39 | 22 | RuntimeError | 2 | 0 | 1 |

Table 10: Distribution of target classes in the Python Runtime Errors dataset.

## B.3 CODE CLASSIFICATION

We present the label distribution for the RunTime error prediction dataset in Table 10. We present the hyper-parameters that we used while finetuning models for code classification tasks in Table 11.

| Hyper-parameters | Ft. linear classification head only | | | Ft. full model end-to-end | | |
|---|---|---|---|---|---|---|
| | Defect | Complexity | Runtime | Defect | Complexity | Runtime |
| Optimizer | AdamW | | | AdamW | | |
| Learning rate (LR) | 1e-3 | | | 5e-5 (baselines) 1e-5 (CODESAGE-SMALL) 1e-5(CODESAGE-BASE) 5e-6(CODESAGE-LARGE) | | |
| LR schedule | Linear | | | Linear | | |
| Batch size | 32 | | | 32 | | |
| # Epoch | 10 | 10 | 2 | 5 | 5 | 2 |

Table 11: Hyperparameters for fine-tuning baseline models and CODESAGE on code classification tasks. Across all models, we used mean pooling to form sequence representations from contextualized token representations.

**Finetuning models end-to-end on classification tasks**   In the main body of this paper, we presented the evaluation results (in Table 2) of finetuning a linear classification head on top of the frozen code representation learning encoders. Furthermore, we finetune the code encoder models end-to-end on the classification tasks and present the results in Table 12. It's evident from these results that CODESAGE outperforms the baseline models.

| Model | Classification | | |
|---|---|---|---|
| | Defect | Complexity | RunTime |
| CodeBERT | $64.37_{0.37}$ | $85.81_{2.53}$ | $42.08_{2.49}$ |
| GraphCodeBERT | $65.36_{1.00}$ | $87.98_{2.45}$ | $44.29_{0.97}$ |
| StarEncoder | $65.20_{0.11}$ | $92.87_{1.47}$ | $38.06_{3.86}$ |
| CodeT5+ Embedding | $64.72_{0.65}$ | $90.63_{1.47}$ | $38.36_{2.54}$ |
| UnixCoder | $65.74_{1.00}$ | $93.75_{0.67}$ | $47.14_{2.71}$ |
| CODESAGE-SMALL | $66.14_{0.67}$ | $94.74_{0.29}$ | $44.46_{1.50}$ |
| CODESAGE-BASE | $\mathbf{66.52}_{0.48}$ | $95.90_{0.43}$ | $46.40_{2.90}$ |
| CODESAGE-LARGE | $66.38_{0.23}$ | $\mathbf{96.20}_{0.57}$ | $\mathbf{49.25}_{3.68}$ |

Table 12: F1 (macro) score of the code classification tasks in the full finetuning setup. We finetuned using three seeds and reported the mean and standard deviation (in subscript).

