# OpenReview forum: "CODE REPRESENTATION LEARNING AT SCALE"
_ICLR.cc/2024/Conference — ICLR 2024 poster_

### Official Review · Reviewer_n8qP · 2023-10-29

**Soundness:** 3 good
**Presentation:** 2 fair
**Contribution:** 2 fair
**Rating:** 6
**Confidence:** 5

**Summary:**

This paper introduces a novel two-step pretraining methodology for encoder-only code language models (LMs). The approach starts with masked language modeling (MLM) and follows up with contrastive learning to foster robust code representations that can be effectively employed in downstream tasks such as retrieval or classification. The resulting model consistently outshines existing baselines across various downstream tasks, including notable ones from the widely recognized CodeXGLUE benchmark. Furthermore, comprehensive ablation studies are conducted to dissect and understand the individual contributions of each component within the proposed framework.

**Strengths:**

1. Code representation learning is a significant and often overlooked aspect of this domain.

2. The evaluation methodology used in this study is solid, showing the proposed method's efficacy through extensive experiments on various downstream tasks from the well-established CodeXGLUE benchmark. Notably, the method exhibits remarkable performance, substantially outperforming established baselines such as CodeT5 and StarEncoder.

3. The inclusion of detailed ablation studies is a strong point of the paper, shedding light on the impact and effectiveness of each component within the proposed method. The insights gleaned from these analyses, especially concerning the methodologies used for contrastive learning, are indeed meaningful.

**Weaknesses:**

1. The naming convention for the model configurations is somewhat confusing. For instance, "CodeSage-Small" (6 layers, 1024 dim, and 130M params) is roughly equivalent to BERT-Base (12 layers, 768 dim, 110M params) in terms of its parameters, while "CodeSage-**Base**" is on par with BERT-**Large** (24 layers, 1024 dim, 330M params). This inconsistency could potentially lead to confusion when making comparisons.

2. The LaTeX typesetting requires significant improvement, as there are inconsistencies in font styles and sizes throughout the paper, notably in Section 4, Figures 2(a) and 6, and Appendix A.5. Additionally, the small text in Figure 4(b) is difficult to read without zooming in, and there are redundant references for CodeBERT and CodeXGLUE.

3. The paper's title, "Code Representation Learning at Scale," is rather broad and lacks specificity.

4. The evaluation could be more comprehensive by including additional tasks, such as POJ104 from the CodeXGLUE benchmark, which is relevant but absent from the current evaluation.

5. The baselines utilized for comparison are generally smaller than the proposed model. It would be beneficial to include larger baselines, such as CodeT5+ (16B) and CodeLLaMa (34B), for a more balanced comparison.

**Questions:**

1. Is there a reason why some tasks from the CodeXGLUE benchmark, like POJ104, were excluded from the evaluation?

---

> ### Author Response · Authors · 2023-11-17
> **Response to Reviewer n8qP**
>
> Thank you for acknowledging the novelty and impact of our work, especially our efforts on diving deep into the key ingredients for code representation learning. We are also grateful for your constructive suggestions on additional benchmarks and baselines, and we have carefully addressed them in our general response (2/2) above. To complement the discussion, we also summarize our initial motivations for experiment design in our general response (1/2).
>
> **Naming Convention**
> It was a hard decision for us. On the one hand, we did want to follow the BERT convention, while we also noticed the recent trend to name the 1B model as the Large model on the [MTEB Leaderboard](https://huggingface.co/spaces/mteb/leaderboard).  However, we are seriously considering your question and trying to find a sweet spot between our current names and CodeSage-Base, CodeSage-Large, and CodeSage-XLarge.
>
> **Latex Typesetting and Paper Title**
> We will improve the formatting as you suggested. We also want to talk about our motivations for the current paper title.
>
> 1. Our work is among the first (in addition to StarCoder, to the best of our knowledge) to train the encoder-based embedding model at a large scale of pretraining data and increased model size. We limit our model size to 1B to balance performance and inference cost in many practical settings, e.g., building a retrieval system over billions of examples.
> 2. More can be different. As reported in Figure 3b of our paper, the dropout-based contrastive learning does not work well with large-scale pretraining data, though it works for UnixCoder and SimCSE, where less than 3M examples are used for training.
> 3. We indeed hope our work can motivate more researchers to keep pushing the frontiers of code embedding models by advancing the pretraining strategy on a large scale of data, as what has been doing for text embeddings ([MTEB](https://huggingface.co/spaces/mteb/leaderboard)).
>
> We are looking forward to hearing your thoughts and would appreciate it if you have better suggestions.
>
> **The Baseline Models Are Smaller**
> One of our goals is to study if larger models would be effective for code representation learning since such models are missing in the literature. Therefore, we urge not to consider that as a weakness of this work.  On the other hand, larger decoders are often not primarily trained to serve retrieval or code understanding works. Those models often yield poor performance in our preliminary evaluations, and we exclude them from comparison as we think it’s unfair to compare with them. However, we appreciate your question, which makes us realize that including these baselines would have made our paper more complete. We summarized the evaluation of CodeGen2.5 (7B), Starcoder (15.5B), and CodeT5+ (16B) in Tables (1a)&(1b) in our general response (2/2) above.
>
> **Evaluation on Clone Detection**
> We did evaluate BigCloneBenchmark at the beginning of this project. However, we found the task very easy but inefficient to evaluate due to the large data size. Hence, we excluded it and focused on more challenging ones. Presumably, given that CodeSage shows strong performances over previous SOTA models on our chosen tasks across many languages, it will perform on par if not able to largely outperform the baselines. Instead, we dived deep and performed analysis to show the reasons behind the effectiveness of CodeSage (e.g., the effectiveness of MLM/DOBF and CL, performance with scaling), which can drive future research works.
>
> But we are grateful for the suggestion, which motivates us to improve our paper's completeness and overall quality. We summarized our evaluation results on both POJ-104 and BigCloneBenchmark in Table 3 in our general response (2/2) above, where CodeSage-small performs slightly better than UniXCoder.
>
> We sincerely thank you for the constructive feedback. We have taken your comments into serious consideration. We provided the evaluation results in our general response (2/2) accordingly, which we will also use for the next update of our paper to address your questions comprehensively. Let us know if you have any other comments.

---

### Official Review · Reviewer_RZCJ · 2023-10-31

**Soundness:** 3 good
**Presentation:** 2 fair
**Contribution:** 1 poor
**Rating:** 3
**Confidence:** 3

**Summary:**

The paper considers the problem of learning better representations of code. The authors propose a model called CodeSage which is based on the Transformer architecture trained on a large corpus of code (The Stack) with various techniques such as masked language modeling, deobfuscation pre-training, and bimodal contrastive learning with hard negative and hard positive examples.

The authors showed that the learned representations can be used for various downstream tasks such as zero-shot code-to-code, natural language-to-code search and code classification. They compared the performance of the proposed model with other prior models such as CodeBERT, GraphCodeBERT, StarEncoder, CodeT5, UniXcoder, and OpenAI's text-embedding-ada-002. The results show that the proposed model outperforms all the other models on nearly all the tasks except for the code classification on code defect detection benchmark.

**Strengths:**

- The paper is well written and the proposed model is well motivated.
- The experiments are thorough and the results are convincing.
- The paper is a good contribution to the field of code representation learning.

**Weaknesses:**

- The paper is not very clear about the differences between the proposed model and the prior models.
- For instance, it seems that the proposed model is a combination of the prior models (CodeBERT and DOBF) with some additional techniques, such as bimodal contrastive learning. It would be good to clarify the differences between the proposed model and the prior models.
- page 3, line 27: remove the last comma in the following square brackets: [x_1, x_2, \ldots, x_N ,]

**Questions:**

How would you explain why the performance of smaller models (e.g., CodeSage-Small and CodeSage-Base) is sometimes better than larger model? Does this mean that the proposed idea is not scalable to larger models or more effective for smaller models?

---

> ### Author Response · Authors · 2023-11-17
> **Response to Reviewer RZCJ**
>
> Thank you for the feedback. To address your main question, we briefly summarize the fundamental contributions of our work below, which differentiate CodeSage from the existing work.
>
> 1. We are the first to train different-sized code embedding models using a much larger amount of publicly available data and study their effectiveness as an off-the-shelf encoder model in downstream tasks.
> 2. We innovate an effective two-stage pretraining strategy in the presence of large-scale data, which outperforms the previous SOTA model by significant margins on a wide range of tasks. In particular,
>      * we propose an effective MLM strategy for code, which is different from the existing literature (including CodeBERT and many others) that mainly follows 80-10-10 corruption convention proposed for text in BERT. We dive deep into why 80-10-10 is suboptimal for code through quantitive and qualitative experiments in Figure 2.
>     *  we identify the value of the DOBF objective that is not utilized in prior works for training code embedding models and cross the technical hurdles to make it work for training encoders (see Appendix A.5 for details). We further boost the performance by leveraging our proposed MLM with DOBF to complement each other (see Table 3 in our paper).
>     * we proposed an effective contrastive learning strategy with effective hard positive construction tricks that boost the performance over the baseline (see Figure 3a).
>     * we deep dive into the key ingredients of code representation learning through an exhaustive ablation study, which we believe will help future works advance the performance further.
>
>
> **Differences Between CodeSage and The Combination of CodeBERT and DOBF**
>
> Our work is not a combination of CodeBERT and DOBF. CodeBERT adopted the original MLM objective proposed for text in the BERT paper. In contrast, we identified the caveat of such naive adoption for code and proposed an effective MLM variant, effectively boosting the performance (see Figure 2).  We are the first to identify the value of the DOBF objective that is not utilized in prior works for training code embedding models and cross the technical challenges to make it work for training encoders (see Appendix A.5). Moreover, we also innovate an effective contrastive learning strategy for stage-II pretraining of our two-stage pretraining scheme. We run exhaustive ablations to show why such two-stage pretraining is necessary and share our findings on the key ingredients for code representation learning.
>
> **Why Smaller Models Sometimes Outperform A Larger Model?**
>
> Large models should generally outperform smaller ones on most tasks; we observe such a trend for CodeSage, especially in the zero-shot settings. However, there are no guarantees that larger models are universally better than smaller ones. We can also clearly see this on the [MTEB leaderboard](https://huggingface.co/spaces/mteb/leaderboard) for text; for example, bge-large-en-v1.5 outperforms bge-base-en-v1.5 (https://huggingface.co/BAAI/bge-large-en-v1.5) on average, but the opposite is also reported on some classification tasks and retrieval tasks, with similar findings for e5-large vs. e5-base (https://huggingface.co/intfloat/e5-large) and many other models.
>
> We have taken your comments into serious consideration and hope our response can help address your question about the contributions of our work. We would be happy to take your other questions, if there are any.

---

### Official Review · Reviewer_iHpZ · 2023-11-02

**Soundness:** 3 good
**Presentation:** 3 good
**Contribution:** 3 good
**Rating:** 6
**Confidence:** 4

**Summary:**

The authors propose pre-training of encoder style for the domain of programming languages at a larger scale (pre-training data and model parameters) than most encoder style models. These models are then tested on tasks in the code understanding domain like text to code search (or semantic code search) and classification tasks on code. The authors also study the impact of different pre-training choices on the model's performance through their ablations to back their methodology.

**Strengths:**

- Scaling the pre-training stage of an encoder only model is a very interesting idea, and hasn't received much attention compared to decoder only models that support generation tasks.

- Exhaustive ablation studies that justify the need for different objectives proposed in pre-training.

- Impressive performance in zero-shot settings on search tasks compared to previous encoder style models

**Weaknesses:**

- Missing comparison with decoder only models: Proposing to scale encoder style models is interesting, but the current results are not convincing as to why this is needed when large scale decoder style pre-trained models have been proposed. While pre-trained decoder only models like CodeGen2.5, CodeLlama, Llama are not specifically proposed for code search or classification, they could be adapted for all the downstream tasks studied by the authors in Sec 4.1. For instance, the last token's encoding from a decoder can be used for search related tasks. Fine-tuning for classification can be performed by framing the classification as a seq2seq problem or by training a classification head on top of the last token's representation. Zero-shot or few shot prompting performance of these decoder style models can be reported for comparison.

- Common Code Understanding benchmarks ignored: The choice of benchmarks is consistent with contemporary literature for the text to code search task, but not for other tasks in the classification settings. This makes it hard for a reader to assess the quality of the pre-trained CodeSage model and compare it to other pre-training strategies from previous works. There are many code understanding tasks (eg. ones from the CodeXGlue benchmark) which have been used in previous works (CodeBERT, GraphCodeBERT, UniXCoder, CodeT5) that are not a part of the evaluation suite here. See Table 1 of https://arxiv.org/abs/2203.03850 - Clone Detection datasets.

- Missing fine-tuned results on CSN, AdvTest and CoSQA: While the benchmark is consistent with previous work for text to code search, the setting chosen is zero shot whereas many previous works show fine-tuned results on the sizeable training datasets provided by these benchmarks (CodeSearchNet, CoSQA, AdvTest). Given that the size of these models is reasonable (130M - 1.3B params) compared to the models studied today (>6-10B params), fine-tuning is an important aspect to be studied with these models.

For instance, MRR scores in Table 2 on the CodeSearchNet benchmark are reported only in the zero-shot setting.  Fine-tuning improves many of these baselines substantially, and beats CodeSage's zero shot performance by significant margin. Post fine-tuning CodeT5+ is at 77 MRR overall, while CodeT5 is at 71, UniXCoder is at 74.4. See Table 6 of https://arxiv.org/abs/2305.07922. Same is the case with CoSQA and AdvTest results. Also, large scale pre-trained decoder only models CodeGen & Llama2 are missing when evaluating in the zero shot setting. While these are not encoder only models, their embeddings can be leveraged for search related tasks.

Also, it's strange why text-embedding-ada-002 is chosen as a baseline here. OpenAI's cpt-code (https://openai.com/blog/introducing-text-and-code-embeddings) would have been a stronger baseline to compare with instead of text-embedding-ada-002 model for code related search tasks.

Writing:
Absract: unclear what downstream task is, is it gen?
Intro: No discussion of next token pred objective from decoder only or encoder-decoder models

Typos:

Writing:
Absract: unclear what downstream task is, is it gen?
structure --> structural aspect
pretraining schemes decide --> dictate/influence/impact

Intro:
para 1: Don't agree that only encoder models are embedding models. Even Llama, CodeLlama, CodeGen can serve as embedding models.
para 2: make it --> makes it
para 3: representation's discriminative power

CodeT5 appears twice in the bib

**Questions:**

- Why are decoder only models not studied as baselines for the code understanding tasks?

---

> ### Author Response · Authors · 2023-11-17
> **Response to Reviewer iHpZ**
>
> Thank you for acknowledging our work's contribution and efforts in enhancing the zero-shot performance and diving deep into the problem through exhaustive analyses. We also appreciate your questions on additional experiments, which indeed motivate us to make our work more complete. Please see our general response (2/2) above for the evaluation results. We also summarized some key motivations of our current experiment design in (1/2) to complement the discussion.
>
> **Missing Comparison with Decoder-only Models**
> As mentioned in our general response (1/2) #1, we tried to avoid unfair comparisons against the larger decoder-only models as they are not primarily trained for discriminative tasks and often yield suboptimal performance. As we can see in Tables (1a)&(1b) in our general response (2/2), CodeGen2.5 (7B), StarCoder (15.5B), and CodeT5+ (16B)-encoder all perform very suboptimal in the zero-shot setting, especially on NL-to-Code search tasks which often require properly designed training objective (e.g., contrastive learning) on the bimodal-data (text-code) to boost the performance.  However, we appreciate your question, which makes us realize we should include these results in our paper for completeness.
>
> **Why Not Code-CPT-001**
> We chose Text-Ada002 models per OpenAI’s suggestion on their website and claim that Ada002 is better or at least comparable to CPT-001 on both text and code tasks but at a much cheaper cost. We have validated their claim in tables (1a) and (1b) in our general response (2/2) above.
>
> **Some Classification Benchmarks Ignored**
> Instead of going through all benchmarks in CodeXGLUE, we preferred to dive deep and perform analysis to show why CodeSage learns effective code representations. We initially evaluated clone detection, which we found is easy for different models but has high evaluation cost due to the large data size. We excluded it and focused on other challenging classification tasks.  However, we have seriously considered your comments and summarized evaluations on POJ-104 and BigCloneBenchmark (BCB) in Table 2 in our general response (2/2). We can see that CodeSage maintains its strong performance.
>
> **Missing Fine-tuned Results on CSN, AdvTest, and CoSQA**
> Our primary goal was to build an off-the-shelf embedding model that does not require finetuning in many practical use cases; semantic search is one such case. Strong zero-shot performance would allow wider usage of the embedding models, as annotations for code are rare and very costly to collect. Our evaluation strategy also aligns with the recent trend in the text domain, where strong models have been actively developed to advance the [MTEB Benchmarks](https://huggingface.co/spaces/mteb/leaderboard). We hope our work can motivate a similar trend for code.
>
> However, we are grateful for your question, which motivates us to better connect with the previous work. We summarized the finetuning results in Table 2 in our general response (2/2), where CodeSage-base (356M) consistently outperforms CodeT5+(770M).  We want to point out that both CodeT5+ 220M and 770M optimize an additional “matching loss” during finetuning (cross-encoder type finetuning of the decoder with pairs selected by the encoder). We hypothesize that if we finetune a copy of UnixCoder or CodeSage on such matching tasks and use them with the original encoders, we can also get a further performance boost for both models.
>
> **Unclear Writing**
> Thank you for pointing out the limitations of the abstract and introduction. We will work on them to clarify our writing in the camera-ready version.
>
> We appreciate the opportunity to improve our paper based on your reviews. We will include these evaluation results reported in our general response above in the next update of our paper to address your questions comprehensively. Let us know if you have any other comments.

---

> > ### Comment · Reviewer_iHpZ · 2023-11-22
> > **Response to rebuttal**
> >
> > Thank you for your response addressing most of the concerns. I appreciate the thoroughness and additional experimental results provided in the rebuttal. I'm increasing my rating of this work to 6.

---

> > > ### Author Response · Authors · 2023-11-22
> > > **Thank You!**
> > >
> > > Thank you for your meticulous and careful review of our manuscript, which motivates us to enhance the completeness and overall quality of our work.  Your thoughtful engagement and willingness to reconsider your review following our rebuttal are truly appreciated.

---

### Official Review · Reviewer_zUhF · 2023-11-08

**Soundness:** 3 good
**Presentation:** 4 excellent
**Contribution:** 4 excellent
**Rating:** 8
**Confidence:** 4

**Summary:**

This paper proposes CodeSage, a new series of models for learning representations of source code, ranging from 130M to 1.3B parameters. CodeSage is pretrained on the Stack dataset, with masked language modeling, identifier deobfuscation, and contrastive learning objectives, and all three objectives are carefully tailored specifically for code representation learning. Experiments are conducted on code search and classification tasks, against strong baselines. Results show that CodeSage not only outperforms all previous open-source code embedding models of similar sizes (i.e., ~125M), scaling it up to 1.3B further improves the performance and it also outperforms the `text-embedding-ada-002` embedding model from OpenAI.

**Strengths:**

S1: All the pretraining objectives are specifically designed for code, the discussions on why such objectives need to be changed (e.g., no random token replacement) for code are very useful for future research in this domain;
S2: To the best of my knowledge, CodeSage 1.3B is the largest (and best performing) code embedding model to date. Scaling up encoder-based models, especially for code representation learning, is an underexplored area;
S3: The experiments are very comprehensive. CodeSage is compared with strong baselines and it was able to beat all of them on the same model size, it also shows that the performance is significantly improved when scaled up to 1.3B. The ablations studies are rather thorough in showcasing the design choices;
S4: The writing of this paper is also great, making it very easy to follow, and conclusions from the figures and tables are very clear. For example, I found Figure 2(a) very helpful in showing why random token replacement makes less sense for code.

**Weaknesses:**

I do not have any significant concerns about this work. But I do have a couple of questions, which are listed in the "Questions" section.

One suggestion: it would be great if we could have an ablation on how each of the training objectives affects the performance (e.g., remove one at a time and see how the performance changes). But I also understand that such ablation is quite expensive.

**Questions:**

Q1: Do you plan to release the model and/or the code?
Q2: About not using random token replacement, this makes a lot of sense to me. But we would probably have the same problem for natural language, if we consider multiple languages, right? Is the same practice common in training multi-lingual text embeddings?
Q3: What is the pretraining hardware and compute?
Q4: Can you comment on the possibility to train even larger code embedding models using this method? It seems that most of the benchmarks are still improving by going from CodeSage-base to CodeSage-large in Table 1 and Table 2.
Q5: For Table 1, are the baseline models also trained on the 9 programming languages being evaluated here?

---

> ### Author Response · Authors · 2023-11-17
> **Response to Reviewer zUhF**
>
> We sincerely thank the reviewer for acknowledging the novelty and impact of our work, especially your great attention to the details we mentioned in the paper.
>
> We also appreciate your constructive suggestions and thoughtful questions. Please see our response below.
>
> **Removing One Objective At A Time**
> We partially cover it through different experiments.
>
> 1. Figure 4a shows that contrastive learning (stage-II pretraining) helps boost performance on top of stage-I pretraining with DOBF & MLM only.
> 2. Figure 5 shows contrastive learning from scratch yields suboptimal performance and cannot benefit from large model sizes. Stage-I pretraining is essential to provide a good starting point for contrastive learning to be effective.
> 3. Table 3 compares DOBF vs. MLM and effective ways to combine them for the stage-I pretraining.
> 4. We initially also tried optimizing all three objectives from scratch. However, we observed suboptimal performance that can be explained by #2 above (which indicates that including contrastive learning from scratch is not effective).
>
> We would like to explore further if we can get computation support for the camera-ready version.
>
> **Responses to Specific Questions**
>
> **Q1:** We plan to release the model checkpoints and the evaluation code.
>
> **Q2:** Your insights align with our hypothesis that random token replacement could hurt MLM training for multilingual text. However, this requires empirical validation; to the best of our knowledge, the existing multilingual MLM encoders are trained with the 80-10-10 practice. Another hypothesis is that the 80-10-10 corruption convention is proposed by the BERT paper, which pre-trained on a comparatively smaller dataset (BooksCorpus + Wikipedia En) over 40 epochs. In such a setting, corruption can help avoid overfitting and cause no significant performance drop. However, nowadays, scaling up the pretraining data allows us to reduce the training epochs, and the effectiveness of the 80-10-10 convention can change consequently.
>
> **Q3:** We used 128 A100 GPUs for pretraining CodeSage-Small&Base, which takes 3-5 days to finish the two-stage pretraining. We used 256 A100 GPUs for pretraining CodeSage-Large, which takes roughly six days. We will add those details in Appendix A.2.
>
> **Q4:** We can train even larger models. In this work, we limit the model size to 1B by considering the cost and latency for many practical settings, e.g., providing embeddings for building a retrieval system over billions of examples. However, if we can get enough support on resources, we are also interested in seeing how the performance can be further improved by scaling up both the data and model size, as we can first optimize for performance and then inference cost.
>
> **Q5:** CodeBERT and GraphCodeBERT are trained on 6 languages, UniXCoder (we used UnixCoder-Nine in this paper) and CodeT5+ embedding models are trained on 9 languages, Starencoder is trained on 86 languages. It is unknown how many programming languages are used to train text-embedding-ada-002.

---

> > ### Comment · Reviewer_zUhF · 2023-11-21
> > **Thanks for you response**
> >
> > I'd like to thank the authors for the details response and the extra experiment results in the general response.
> >
> > I found the clarifications and answers to my questions clear and helpful. And I am glad to know that there are plans to release the model checkpoints and evaluation harness.
> >
> > I stand by my original assessment and will keep supporting this work. I will also keep monitoring the discussions with the other reviewers.

---

> > > ### Author Response · Authors · 2023-11-21
> > > **Thank you!**
> > >
> > > We sincerely want to express our deepest gratitude for the time and effort Reviewer zUhF dedicated to thoroughly reading our paper and the accompanying rebuttal. Your recognition of our work is not only encouraging but also greatly motivates us for further explorations along this direction.  Thank you once again.

---

### Author Response · Authors · 2023-11-17
**General Response by Authors (1/2)**

We thank the reviewers for your thoughtful comments and constructive suggestions. We are pleased that most reviewers acknowledged the impact of our work, especially those comments that our method is novel, our model performance is strong, and our experiments are exhaustive.

We are also grateful for your comments and questions on additional evaluations. We have considered and carefully addressed them in part two of this response. Below, we'd like to summarize a few key points of our initial motivation on experiment design to complement the discussion.

1. **Model Size and Large Decoders**  One of the primary use cases we consider is retrieval, which can require embedding billions of examples in many practical applications. We limit our embedding model size to 1B for affordable efficiency and cost. However, we didn’t compare with the decoder models mainly because of their poor performance on the zero-shot tasks in our preliminary evaluations, which is expected as they are not primarily trained for discriminative tasks. Hence, the comparison can be unfair, in our opinion.

2. **Why Zero-Shot** Creating a large training corpus is often time-consuming and expensive, especially for code. We primarily focus on bridging the gap between zero-shot and finetuning by innovating an effective pretraining strategy. Notably, CodeSage-small outperforms the previous SOTA model with a similar size by 18% to 51% relative improvement on the NL2Code search and 21.6% on the Code2Code search.  Further, zero-shot evaluation or light-finetuning of a linear layer provides a good measurement of the representation quality, which aligns with the recent trend that various text embedding models have been actively developed to advance the [MTEB benchmarks](https://huggingface.co/spaces/mteb/leaderboard).

3. **Why Some Classification Tasks Are Ignored** Instead of going through all CodeXGLUE tasks, we prioritized diving deep into the key ingredients for code representation learning through exhaustive ablations. We initially considered the Clone-Detection with BigCloneBenchmark, which we found is very easy for models but has a high evaluation cost due to the large data size. We ignored them and focused on the more challenging classification tasks presented in our paper.

4. **Text-Ada-002 vs. Code-CPT-001** We select the OpenAI-Text-Ada-002 model per OpenAI’s suggestion on their website and claim that Ada-002 is better or at least comparable to CPT-001 on both text and code tasks, but with much lower cost (it took us $300+ to get the embeddings from Babbage-Code-001 for the evaluations in Tables 1a&1b in part two of this response). However, we appreciate the question, which motivates us to validate their claim.

We summarized the evaluation results in the follow part. We appreciate the opportunity to add these results in the next update of our paper based on your feedback to improve our work's completeness and overall quality.

---

> ### Author Response · Authors · 2023-11-17
> **General Response by Authors (2/2)**
>
> **Benchmarking with Large Decoder Models && OpenAI Ada-002 vs Code-CPT-001**
>
> 1. Decoders, even with large model sizes, often perform poorly on discriminative tasks in the zero-shot setting, as they are not primarily trained for those tasks. In particular, NL2Code search often requires properly designed objectives on the bimodal text-code data, e.g., contrastive learning or conditional generation.
> 2. OpenAI-Code-CPT-001 underperforms Text-Ada-002 on code tasks, which aligns with their claim.
>
> * ***Table (1a)*** Zero-Shot Code-to-Code Search
>
> | Model                | Python | Java |  JS |  TS |  C# |  C  | Ruby | PHP  | GO  | Avg  |
> |----------------------|--------|------|-----|-----|-----|-----|------|------|-----|------|
> | CodeGen2.5(7B)       | 16.5   | 10.2 | 7   | 8.5 | 4.2 | 8.0 | 17.3 | 15.6 | 9.4 | 10.7 |
> | Starcoder(15.5B)     | 7.1    | 3.9  | 3.2 | 4.4 | 1.7 | 2.4 | 6.8  | 6.1  | 3.3 | 4.3  |
> | CodeT5+(16B) Encoder | 18.2   | 9.9  | 5.8 | 6.9 | 4.2 | 8.2 | 16.5 | 13.9 | 8.0 | 10.2 |
> | Babbage-code-001(CPT)| 21.9   | 8.9  | 4.9 | 5.7 | 3.2 | 11.6| 26.3 | 16.6 | 9.4 | 12.0 |
> | OpenAI-Text-Ada002   |*35.9*  |*25.1*|*19.0*|*21.9*|*10.2*|*29.2*|*40.9*|*40.5*|*23.4*|*27.3*|
>
> * ***Table (1b)*** Zero-Shot NL-to-Code Search
>
> | Model                 | CSN-Py | CSN-Java | CSN-JS | CSN-PHP | CSN-GO | CSN-Ruby | CoSQA | AdvTest |
> |-----------------------|--------|----------|--------|---------|--------|----------|-------|---------|
> | CodeGen2.5 (7B)       |  0.02  |   0.01   |  0.06  |  0.02   |  0.05  |   0.18   | 6.03  | 2.04    |
> | Starcoder (15.5B)     |  0.02  |   0.06   |  0.03  |  0.01   |  0.05  |   0.59   | 0.06  | 0.05    |
> | CodeT5+ (16B) Encoder |  22.96 |   20.36  |  19.93 |  14.05  |  12.26 |   26.08  | 20.37 | 13.05   |
> | Babbage-code-001 (CPT)|  63.13 |  67.85   |  62.30 |  57.47  |  85.22 |   69.28  |*52.20*| 36.03   |
> | OpenAI-Text-Ada-002   |*68.02*| *71.49*  | *67.50*| *60.62* | *85.63*|  *74.20* | 44.23 |*38.08*  |
>
> **Finetuning Results on NL2Code Search**
>
> CodeSage maintains the performance gain over the baseline models, especially that CodeSage-base (356M) persistently outperforms CodeT5+(770M). We want to point out that both CodeT5+ 220M and 770M optimize an additional “matching loss” during finetuning (cross-encoder type finetuning of the decoder with pairs selected by the encoder). We hypothesize that if we finetune a copy of UnixCoder or CodeSage on such matching tasks and use them with the original encoders, both models can get a further performance boost.  (Note: Finetuning CodeSage-Large is undergoing; we will add the results once it is complete.)
>
> * ***Table 2***  Supervised Finetuning on NL2Code Search
>
> | Model               | CSN-Ruby | CSN-JS | CSN-GO | CSN-Py | CSN-Java |CSN-PHP | CSN-Avg | CoSQA | AdvTest |
> |---------------------|----------|--------|--------|--------|----------|--------|---------|-------|---------|
> | UnixCoder(125)      |   74.0   |  68.4  |  91.5  | 72.0   |   72.6   |  67.6  |  74.4   | 70.1  | 41.4    |
> | CodeT5 (220M)       |   71.9   |  65.5  |  88.8  | 69.8   |   68.6   |  64.5  |  71.5   | 47.8  | 39.3    |
> | CodeT5+(220M)       |   77.7   |  70.8  |  92.4  | 75.6   |   76.1   |  69.8  |  77.1   | 72.7  | 43.3    |
> | CodeT5+(770M)       |   78.0   |  71.3  |  92.7  | 75.8   |   76.2   |  70.1  |  77.4   | 74.0  | 44.7    |
> | CodeSage-small(130M)|   76.2   |  69.7  |  91.6  | 73.0   |   72.8   |  67.1  |  75.1   | 70.2  | 44.7    |
> | CodeSage-base(356M) |   79.2   |  73.3  |  92.6  | 76.8   |   75.9   |  69.8  |**77.9** |**74.2**|**53.0**|
>
>
>
> **Additional Benchmarking Results on Code Clone Tasks**
>
> We report the finetuning results by using the evaluation code from UnixCoder. We are working on the evaluations for other baseline models and will add them once they are ready.
>
> * ***Table 3*** Supervised Finetuning on POJ-104 and BCB
>
> | Model          | POJ-104 MAP@R | BCB (Recall) | BCB (Precision) | BCB (F1) |
> |----------------|---------------|--------------|-----------------|----------|
> | UniXCoder      | 90.52         | 92.9         | 97.6            | 95.2     |
> | CodeSage-small | 90.72        | 95.1         | 95.9            | 95.5     |
> | CodeSage-base  | 92.37         | 94.9         | 96.3            | 96.1     |
> | CodeSage-large | 94.14         | 96.1         | 96.9            | 96.5     |

---

### Meta-Review · Area_Chair_jhAT · 2023-11-24

**Metareview:**

This work presents a specialized code encoder model trained at scale. This is achieved by using a combination of DOBF, random masking, and contrastive learning. Evaluation shows good improvements compared to non-code-specific models.

### Strengths:
- Simple and clear training method.
- Good empirical improvements across tasks.
- Comprehensive evaluation.

### Weaknesses
- Limited novelty in terms of modeling.

Given the above, I believe that this work should be accepted.

##### Other comments
- I’ve been unable to find in the text how the single embedding is computed from the transformer encoder outputs. I imagine that this is the representation of the `[CLS]`-like token? (For example, pooling $h_i$ could have also been used). Please clarify in the text.
- A single vector representation for a code snippet seems a heavily lossy compression. How does performance deteriorate wrt to the length of the code snippet?

**Justification For Why Not Higher Score:**

While there is no reason to reject this work, the performance improvements are relevant to a relatively small part of the community and the novelty of this work is not spotlight-worthy.

**Justification For Why Not Lower Score:**

Rejecting this work would deprive the community of information about improving code encoders, which can find many applications.

---

### Decision · Program_Chairs · 2024-01-16

Accept (poster)